# Generative Risk Minimization for Out-of-Distribution Generalization on Graphs

**Song Wang**                                                                     *sw3wv@virginia.edu*
*Department of Electrical and Computer Engineering*
*University of Virginia*

**Zhen Tan**                                                                         *ztan36@asu.edu*
*Department of Electrical and Computer Engineering*
*University of Virginia*

**Yaochen Zhu**                                                                   *uqp4qh@virginia.edu*
*Department of Electrical and Computer Engineering*
*University of Virginia*

**Chuxu Zhang**                                                              *chuxu.zhang@uconn.edu*
*School of Computing*
*University of Connecticut*

**Jundong Li**                                                                    *jundong@virginia.edu*
*Department of Electrical and Computer Engineering*
*University of Virginia*

**Reviewed on OpenReview:** `https://openreview.net/forum?id=EcMVskXo1n`

## Abstract

Out-of-distribution (OOD) generalization on graphs aims at dealing with scenarios where the test graph distribution differs from the training graph distributions. Compared to i.i.d. data like images, the OOD generalization problem on graph-structured data remains challenging due to the non-i.i.d. property and complex structural information on graphs. Recently, several works on graph OOD generalization have explored extracting invariant subgraphs that share crucial classification information across different distributions. Nevertheless, such a strategy could be suboptimal for entirely capturing the invariant information, as the extraction of discrete structures could potentially lead to the loss of invariant information or the involvement of spurious information. In this paper, we propose an innovative framework, named Generative Risk Minimization (GRM), designed to *generate* an invariant subgraph for each input graph to be classified, instead of extraction. To address the challenge of optimization in the absence of optimal invariant subgraphs (i.e., ground truths), we derive a tractable form of the proposed GRM objective by introducing a latent causal variable, and its effectiveness is validated by our theoretical analysis. We further conduct extensive experiments across a variety of real-world graph datasets for both node-level and graph-level OOD generalization, and the results demonstrate the superiority of our framework GRM. Our code is provided at https://github.com/SongW-SW/GRM.

## 1 Introduction

In recent years, it has become increasingly crucial to develop machine learning models that can handle tasks with test data distributions differing from training data, commonly referred to as out-of-distribution (OOD) generalization (Mansour et al., 2009; Blanchard et al., 2011; Muandet et al., 2013; Beery et al., 2018; Recht et al., 2019; Su et al., 2019). Such disparities, termed as *distribution shifts*, can substantially undermine the

efficacy of the empirical risk minimization (ERM) paradigm, which presumes consistency in data distribution across training and test phases (Quinonero-Candela et al., 2008; Lazer et al., 2014; Zhang et al., 2018). While there exist numerous works (Hu et al., 2018; Krueger et al., 2021; Chang et al., 2020; Sagawa et al., 2020; Koh et al., 2021) on OOD generalization for i.i.d. (independent and identically distributed) data (e.g., images), few have focused on graph-structured data, despite the prevalence of distribution shifts in real-world graphs (Fakhraei et al., 2015; Gui et al., 2022; Yu et al., 2023). For instance, in citation networks (Hu et al., 2020), the distribution of paper topics (i.e., node labels) may considerably change over time, leading to differences in graph structures. However, Graph Neural Networks (GNNs) (Kipf & Welling, 2017; Hamilton et al., 2017; Xu et al., 2019; Zhou et al., 2020; You et al., 2020), despite being the de facto choice to model graphs, often fall short in addressing the challenge of distribution shifts on OOD graph data (AlBadawy et al., 2018; Dai & Van Gool, 2018; Li et al., 2022a; Tan et al., 2022a; Zhang et al., 2024).

Existing works for OOD generalization primarily focus on identifying *invariant relationships* across diverse data distributions, generally referred to as *environments* (or domains) (Arjovsky et al., 2019; Chang et al., 2020; Ahuja et al., 2020). They typically aim to identify this invariant relationship, which maps input invariant features to the outputs (e.g., labels), through robust optimization or learning an invariant feature space (Creager et al., 2021; Krueger et al., 2021). Regarding OOD generalization on graphs, existing works (Chen et al., 2022; Miao et al., 2022; Chen et al., 2023; Tan et al., 2023; Wang et al., 2024) primarily aim to identify an invariant subgraph $G_c$ from a given graph $G$ for predictions. However, such an extraction strategy could be subpar for completely capturing the invariant information, which could be mixed with spurious information in a graph and could not be distinctly separated (Bevilacqua et al., 2021). For example, due to the complicated interactions (as edges) of atoms (as nodes) in a molecule graph, extracting a node may inevitably incorporate both invariant and spurious information, thus failing to achieve a precise invariant subgraph (Gui et al., 2022). Concretely, the strategy of extracting (discrete) structures may not extract invariant information on graphs.

To deal with this, we propose an innovative framework, named Generative Risk Minimization (GRM), to fully exploit invariant information on graphs. Different from the distinct extraction of invariant subgraphs used in existing works (Chen et al., 2023; 2022), the core idea of GRM is to generate the invariant subgraphs in a continuous manner. In particular, the generated invariant subgraph preserves the same set of nodes as the input $G$, while possessing continuous edge weights and node representations. This design allows us to flexibly preserve the invariant information without the need for extracting discrete structures, which could potentially lead to loss of invariant information. To ensure that the generated subgraphs contain sufficient invariant information, our proposed GRM framework involves two objectives: (1) generation objective, which aims to generate precise subgraphs with continuous edge weights and node representations, and (2) invariant objective, which ensures the independence of the invariant subgraph and the domains.

Although the above two objectives are straightforward and intuitive, it is challenging to directly optimize them, especially when the domain labels are unavailable (Wu et al., 2022b). Therefore, we transform our GRM objective into three correlated losses that could be used for optimization, based on our theoretical analysis. In particular, our GRM framework achieves two attractive properties for OOD generalization. (1) Maximally involving invariant information. We introduce a variational approximation of the latent causal variable $Z$ to both the generation objective and the invariant objective. Our derivation results ensure that the learned latent representation of $Z$ involves minimal loss of invariant information. (2) Minimally involving spurious information. Our GRM framework could directly minimize the mutual information between the invariant subgraph and the domains when combined with our generation objective, based on our theoretical analysis. The derived loss forces the generator to focus less on domain-related information and thereby reduces the incorporation of spurious information. In summary, our contributions are as follows:

- We develop the Generative Risk Minimization (GRM) framework, a novel approach that aims to generate invariant subgraphs for graph OOD generalization.

- We provide a theoretical analysis that ensures the effectiveness of the generated subgraphs and sheds light on the rationales of GRM and its validity in graph OOD generalization tasks.

- We evaluate GRM through extensive experiments on various real-world datasets that cover multiple types of distribution shifts. The results validate the superiority of GRM over state-of-the-art baselines.

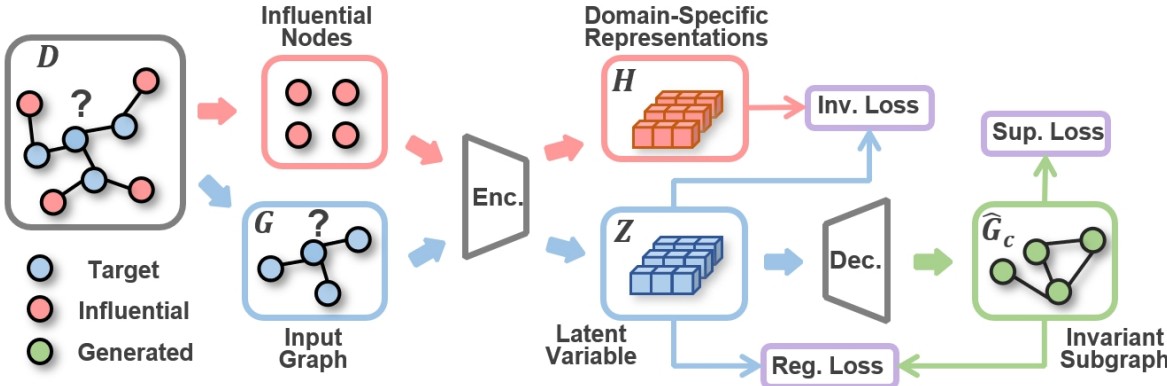

Figure 2: The overall framework of GRM. Each input graph $G$ is processed by the encoder of our generator to learn the latent variable $Z$. Then we extract the most influential nodes from the domain and learn a domain-specific representation for each node in $G$. These domain-specific representations will be used in the invariance loss. We further classify the output invariant subgraph with a classifier to obtain the predictions. The regularization loss is calculated for $Z$ and the invariant subgraph.

## 2 Preliminaries

In this section, we provide the formulation for our studied graph OOD generalization problem. We start by representing a graph (or a local subgraph of a node in node-level tasks) as $G = (\mathcal{V}, \mathcal{E}, \mathbf{X})$, where $\mathcal{V}$ and $\mathcal{E}$ are the node set and the edge set, respectively. Moreover, $\mathbf{X} \in \mathbb{R}^{|\mathcal{V}| \times d_x}$ is a feature matrix, where the $j$-th row vector ($d_x$-dimensional) represents the attribute of the $j$-th node. We can define the distribution of a graph and its label from domain $D_i$ as $(G, Y) \sim P(G, Y | D_i)$, where $Y \in \mathcal{Y}$ is the label of $G$. Here $\mathcal{Y}$ is the label space shared across domains. We further denote the training and test domains (i.e., graphs) as $\mathcal{D}_{tr} = \{D_1, D_2, \ldots, D_{|\mathcal{D}_{tr}|}\}$ and

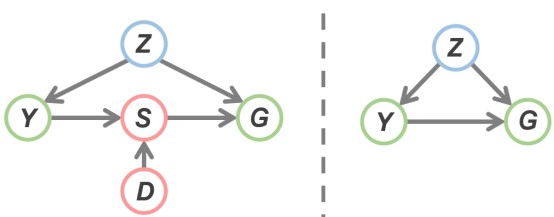

Figure 1: The SCMs with distribution shift (left) and without distribution shifts (right).

$\mathcal{D}_{te} = \{D_1, D_2, \ldots, D_{|\mathcal{D}_{te}|}\}$, respectively. Generally, existing works for OOD generalization on graphs primarily rely on the Structural Causal Models (SCMs), as shown in Fig. 1, to interpret distribution shifts on graphs (Chen et al., 2022; 2023). Specifically, the observed graphs $G$ and labels $Y$ are affected by the latent causal variable $Z$ and a spurious variable $S$, which decide the underlying invariant subgraph $G_c$ and the spurious subgraph $G_s$, respectively. As the spurious variable $S$ is related to the domain $D$, existing works aim to identify the invariant subgraph $G_c$ for the precise prediction of its label $Y$, without the effect of $S$. Specifically, the goal is to develop an invariant GNN, represented as $\mathcal{M} := f_c \circ g$. This model comprises: 1) an extractor $g : \mathcal{G} \to \mathcal{G}_c$ that identifies the invariant subgraph $G_c$, and 2) a classifier $f : \mathcal{G}_c \to \mathcal{Y}$ that predicts the label $Y$ using the extracted $G_c$, where $\mathcal{G}_c$ denotes the space of subgraphs within $G$. However, the extractor $g$ in existing works could only output a discrete invariant subgraph $\widehat{G}_c$, which is a subset of edges and nodes in the input graph $G$. As a result, the invariant information and spurious information cannot be entirely separated when a node or edge consists of both types of information.

## 3 Methodology

In this section, we elaborate on our proposed Generative Risk Minimization (GRM) framework, which aims to tackle the graph OOD generalization problem by generating invariant subgraphs instead of extraction. In the following, we first derive our proposed GRM objective and then introduce specific designs to optimize the objective in a generative manner. The overall process of our GRM framework is illustrated in Fig. 2.

### 3.1 GRM Objective

In our GRM framework, we propose to learn a classifier $f(\cdot)$ and a generator $g(\cdot)$, such that the generator $g(\cdot)$ will output an invariant subgraph for each input graph. Considering a graph input $G = (\mathcal{V}, \mathcal{E}, \mathbf{X})$, the generator aims to outputs the invariant subgraph $\widehat{G}_c = (\widehat{\mathcal{V}}, \widehat{\mathcal{E}}, \widehat{\mathbf{X}})$ for classification.

To ensure that the obtained subgraph is maximally invariant across domains while preserving the causal information, we consider the following learning objective for $f$ and $g$, which is adopted in existing works (Chen et al., 2022; 2023):

$$\max \ I(\widehat{G}_c; Y), \quad \text{s.t.} \quad \widehat{G}_c \perp D, \ \widehat{G}_c = g(G). \tag{1}$$

However, it is difficult to directly optimize this objective. Generally, the optimization objective of the generator is to maximize the log-likelihood term $\log P(\widehat{G}_c|G)$. Combining this term, we propose a more feasible objective for Eq. (1):

$$\max \ \mathbb{E}\left[\log P(\widehat{G}_c|G)\right] - I(\widehat{G}_c; D), \tag{2}$$

which is referred to as our proposed GRM objective. Although the GRM objective is straightforward, it is intractable due to the lack of ground truth, i.e., $G_c$, for the generated invariant subgraph $\widehat{G}_c$. Alternatively, based on the SCMs on distribution shifts as illustrated in Fig. 1, we propose to model the causal variable $Z$ as a latent variable for graph generation. By introducing the latent causal variable $Z$, we are able to derive the following theorem that allows for an end-to-end optimization for our objective in Eq. (2).

**Theorem 3.1.** *An evidence lower bound (ELBO) for optimization of the GRM objective, by introducing a latent causal variable $Z$ and variational approximations $Q(Z)$ and $Q(\widehat{G}_c)$, is as follows:*

$$\max \ \mathbb{E}\left[\log P(\widehat{G}_c|G, Z)\right] - KL(Q(Z)\|P(Z|G)) - \mathbb{E}[KL(P(\widehat{G}_c|D, Z)\|Q(\widehat{G}_c))] + \mathbb{E}\left[\log P(Z|D, \widehat{G}_c)\right]. \tag{3}$$

The proof is provided in Appendix A.1. $KL(\cdot\|\cdot)$ denotes the Kullback-Leibler (KL) divergence. Based on the above objective, we could devise specific losses for optimization of the classifier and generator.

### 3.2 Generator Implementation

Before we derive the detailed optimization losses based on Theorem 3.1, we first introduce the implementation of our generator. In particular, we aim to model $Q(Z)$ using the generator $g$, which uses any graph $G$ as input. However, it remains challenging to model $Z$ with a suitable architecture of the generator $g$ in the absence of the ground truth $\widehat{G}_c$. In particular, we propose to leverage the Variational Graph Auto-Encoder (VGAE) (Kipf & Welling, 2016; Simonovsky & Komodakis, 2018) for the generation of invariant subgraphs. This is because the optimization objective of VGAE involves a latent variable $Z$ and aligns with the first term of the GRM objective. As such, we propose to implement the generator $g(\cdot)$ as a VGAE.

Following the VGAE architecture, our generator consists of an encoder and a decoder. Given a graph input $G$, the encoder maps it into a latent space and outputs the latent variable $Z \in \mathbb{R}^{|\mathcal{V}| \times d_z}$. Here $d_z$ is the dimension size of $Z$. Moreover, $Z$ involves $|\mathcal{V}|$ latent representations, i.e., $Z = \{\mathbf{z}_1, \mathbf{z}_2, \ldots, \mathbf{z}_{|\mathcal{V}|}\}$, which means we learn a latent representation for each node in $G$, and thus the number of nodes in $\widehat{G}_c$ equals that in $G$. For each node $v_i$, where $i \in \{1, 2, \ldots, |\mathcal{V}|\}$, we learn its representation as follows:

$$\mathbf{z}_i \sim \mathcal{N}(\mathbf{z}|\mu_i, \text{diag}(\sigma_i^2)), \quad \text{where } \mu_i = \text{GNN}_\mu(\mathcal{V}, \mathcal{E}, X)_i \text{ and } \log \sigma_i = \text{GNN}_\sigma(\mathcal{V}, \mathcal{E}, X)_i. \tag{4}$$

To generate node features of the invariant subgraph, i.e., $\widehat{\mathbf{X}} \in \mathbb{R}^{|\mathcal{V}| \times d_x}$, we leverage the obtained latent variable $Z$ along with a linear projection layer $f_x(\cdot)$:

$$\widehat{\mathbf{X}} = \{f_x(\mathbf{z}_1), f_x(\mathbf{z}_2), \ldots, f_x(\mathbf{z}_{|\mathcal{V}|})\}, \quad \text{where } f_x(\mathbf{z}_i) = \mathbf{W_x}\mathbf{z}_i + \mathbf{b_x}. \tag{5}$$

Here $\mathbf{W_x} \in \mathbb{R}^{d_x \times d_z}$ is the weight of the projection layer, and $\mathbf{b_x} \in \mathbb{R}^{d_x}$ is the bias. Then we further generate edges from the latent variables $\mathbf{z}_i$ as follows:

$$\widehat{\mathcal{E}} = \{\widehat{e}_{ij}|i, j = 1, 2, \ldots, |\mathcal{V}|\}, \quad \text{where } \widehat{e}_{ij} = \sigma(f_e^\top(\mathbf{z}_i) \cdot f_e(\mathbf{z}_j)) \text{ and } f_e(\mathbf{z}_i) = \mathbf{W_e}\mathbf{z}_i + \mathbf{b_e}. \tag{6}$$

Here $\mathbf{W_e} \in \mathbb{R}^{d_e \times d_z}$ is the weight of the projection layer, and $\mathbf{b_e} \in \mathbb{R}^{d_e}$ is the bias. $d_e$ is the dimension size of $f_e(\mathbf{z}_i)$. $\sigma(x) = 1/(1 + \exp(-x))$ is the sigmoid function. Notably, unlike traditional graph generation tasks (De Cao & Kipf, 2018; Jin et al., 2018), here we keep the continuous values of $\widehat{e}_{ij}$ as the edge weight and do not sample discrete edges. This is because we aim to generate precise subgraphs that maximally preserve the invariant information, and sampling discrete edges could potentially incorporate spurious information or cause the loss of invariant information. Through the above steps, we could generate an invariant subgraph $\widehat{G}_c = (\widehat{\mathcal{V}}, \widehat{\mathcal{E}}, \widehat{\mathbf{X}})$, given an input graph $G$.

## 3.3 Optimization based on GRM

In this subsection, we introduce the detailed process to optimize our framework based on the GRM objective derived in Theorem 3.1. In particular, we design three different losses for the terms in the derivation result.

**Supervision Loss.** For the supervision loss, we first consider the term $\mathbb{E}[\log(P(\widehat{G}_c|G, Z))]$. As the ground truth $G_c$ is unobserved, the common choice of reconstruction loss in VGAE is unavailable. Therefore, we propose to adopt the label $Y$ of $G$ as a proxy for $G_c$, based on the intuition that the optimal $G_c$ should maximally reflect the information of the label $Y$. In this manner, we could formalize the supervision loss as follows:

$$\mathcal{L}_s = -\sum_{y \in \mathcal{Y}} p(y|G) \log p(y|\widehat{G}_c), \text{ where } p(y|\widehat{G}_c) = f_y(\widehat{G}_c) \text{ and } \widehat{G}_c = g(G). \tag{7}$$

In the above loss, $p(y|\widehat{G}_c)$ is obtained by taking $\widehat{G}_c$ as input to the classifier $f(\cdot)$, and $f_y(\cdot)$ denotes the output class probability regarding class $y$. Moreover, we set $p(y|G) = 1$ if $y$ is the label of $G$, and $p(y|G) = 0$, otherwise. The above supervision loss could be interpreted as a cross-entropy classification loss for $\widehat{G}_c$.

**Regularization Loss.** Generally, KL-divergence terms act as regularization in variational generation (Kipf & Welling, 2016; Kingma et al., 2019; Simonovsky & Komodakis, 2018). In our derivation of the GRM objective in Theorem 3.1, the two KL-divergence terms $-\mathrm{KL}(Q(Z)\|P(Z|G))$ and $-\mathrm{KL}(P(\widehat{G}_c|D, Z)\|Q(\widehat{G}_c))$ represent the differences between the distributions of $Z$ (given $D$) and $Q(Z)$, as well as between the distributions of $\widehat{G}_c$ (given $D$ and $Z$) and $Q(\widehat{G}_c)$. Notably, the derived result is applicable for any $Q(Z)$ and $Q(\widehat{G}_c)$. Specifically, we first define $Q(Z)$ as a Gaussian distribution $\mathcal{N}(0, \mathbf{I})$, where $\mathbf{I} \in \mathbb{R}^{d_z \times d_z}$ is the identity matrix. In this way, we could directly regularize the learned $\mu$ and log of $Z$, as $P(Z|G)$ is also a Gaussian distribution, and thus we could explicitly derive the KL-divergence between it and $\mathcal{N}(0, \mathbf{I})$. For the second KL-divergence term, i.e., $-\mathrm{KL}(P(\widehat{G}_c|D, Z)\|Q(\widehat{G}_c))$, we first formulate $Q(\widehat{G}_c)$ as $Q(\widehat{G}_c) = Q(\widehat{\mathbf{X}}) \cdot Q(\widehat{\mathcal{E}})$. In this manner, we could obtain:

$$-\mathrm{KL}(P(\widehat{G}_c|D, Z)\|Q(\widehat{G}_c)) = -\mathrm{KL}(P(\widehat{\mathbf{X}}|D, Z)\|Q(\widehat{\mathbf{X}})) - \mathrm{KL}(P(\widehat{\mathcal{E}}|D, Z)\|Q(\widehat{\mathcal{E}})). \tag{8}$$

Notably, as $\widehat{\mathbf{X}}$ is the linear projection of $Z$, the term $-\mathrm{KL}(P(\widehat{\mathbf{X}}|D, Z)\|Q(\widehat{\mathbf{X}}))$ could also use $Z$ for calculating the regularization loss in a similar way to $-\mathrm{KL}(Q(Z)\|P(Z|G))$. For another term $-\mathrm{KL}(P(\widehat{\mathcal{E}}|D, Z)\|Q(\widehat{\mathcal{E}}))$, we could decompose $Q(\widehat{\mathcal{E}})$ into multiple independent Bernoulli distributions as $\widehat{e}_{ij} \sim \text{Bernoulli}(\theta)$, where $\theta \in [0, 1]$ is a controllable hyper-parameter. In this manner, we could consider the learned edge weight $\widehat{e}_{ij}$ as the parameter in a Bernoulli distribution and compute its KL-divergence with $Q(\widehat{\mathcal{E}})$. In concrete, we formulate the regularization loss as follows:

$$\mathcal{L}_r = \sum_{i=1}^{d_z} \left( \frac{1}{2}(\sigma_i^2 + \mu_i^2) - \log \sigma_i \right) + \sum_{i=1}^{|\mathcal{V}|} \sum_{j=1}^{|\mathcal{V}|} \left( r(\alpha_{ij}, \theta) + r(1 - \alpha_{ij}, 1 - \theta) \right), \tag{9}$$

where $r(\alpha, \theta) = \alpha \log(\alpha/\theta)$. The first term is calculated from the KL-divergence of the two Gaussian distributions, which is $\mathrm{KL}(P(\widehat{\mathbf{X}}|D, Z)\|Q(\widehat{\mathbf{X}}))$. The second term is calculated from KL-divergence between the two Bernoulli distributions, which is $\mathrm{KL}(P(\widehat{\mathcal{E}}|D, Z)\|Q(\widehat{\mathcal{E}}))$. Particularly, this loss regularizes the learning process of latent variable $Z$ and the generation process of invariant subgraph $\widehat{G}_c$, such that the obtained $\widehat{G}_c$ is more generalizable to various domains.

**Invariance Loss.** Finally, we consider the thrid term derived in Theorem 3.1, i.e., $\mathbb{E}[\log P(Z|D, \widehat{G}_c)]$. Intuitively, this term aims to derive the correct latent variable $Z$ given the generated invariant subgraph $\widehat{G}_c$

and domain $D$. However, the ground truth of $Z$ is unavailable during. Thus, we propose to use the latent variable $Z$ generated from $\widehat{G}_c$ in Eq. (4) as the proxy and minimize the discrepancy between $Z$ and another set of latent variable $H = \{\mathbf{h}_1, \mathbf{h}_2, \ldots, \mathbf{h}_{|\mathcal{V}|}\}$ learned from the domain $D$. We refer to $H$ as the domain-specific latent variable. In this manner, optimizing this term could make the learned latent variable $Z$ less vulnerable to the effect of domain-specific information, thereby enhancing the invariance of $Z$. Specifically, we aim to precisely capture the domain information that is maximally related to nodes in $\mathcal{V}$. Due to the diversity of nodes within each domain, the useful domain information can be different for various nodes in $G$ and also distributed across the entire graph (Gui et al., 2022). Therefore, we propose to learn $\mathbf{h}_i$ by considering nodes that are influential on $v_i$. Specifically, we construct a subgraph from these influential nodes for $v_i$ and learn $\mathbf{h}_i$ from this subgraph. To effectively select influential nodes, we consider both the shortest path distance and the number of shortest paths. In practice, we choose the one-hop neighboring node set $\mathcal{N}_i$ of $v_i$ and select nodes that are most influential to $\mathcal{N}_i$ to maximally capture domain information. Notably, we select the one-hop neighboring node set because according to Theorem 1 in (Huang & Zitnik, 2020), the influence of one node on another decreases exponentially as the distance between the two nodes increases. As a result, to determine the nodes from the domain that are most influential for a specific node, it is logical to prioritize nodes with the smallest distances. To summarize, we can represent the selected nodes for learning the domain-specific representation $\mathbf{h}_i$ of node $v_i$ (the $i$-th node in $\mathcal{V}$) as follows:

$$\mathcal{V}_i^D = \{u | \overline{L}_S(u, \mathcal{N}_i) \leq L^*, \widetilde{P_S}(u, \mathcal{N}_i) \geq P^*, \}, \text{ where } i = 1, 2, \ldots, |\mathcal{V}|. \tag{10}$$

Here $L^* \in \mathbb{R}$ and $P^* \in \mathbb{R}$ are hyper-parameters that control the number of selected nodes for learning domain representations based on $\overline{L}_S$ and $\widetilde{P_S}$, respectively. $\mathcal{N}_i$ is the set of one-hop neighboring nodes of $v_i$, and $\mathcal{V}$ is the node set of $G$. In this way, we can learn the domain-specific representation $\mathbf{h}_i$ of node $v_i$ as follows:

$$\mathbf{h}_i = \text{ Mean} \left( \text{GNN}_\mu(\mathbf{X}_i^D, \mathcal{V}_i^D, \mathcal{E}_i^D) \right), \text{ where } \mathcal{E}_i^D = \{(v_a, v_b) | v_a, v_b \in \mathcal{V}_i^D\}. \tag{11}$$

Here, $\mathbf{X}_i^D$ and $\mathcal{E}_i^D$ are the corresponding node features and edge set of $\mathcal{V}_i^D$, respectively. $\mathbf{h}_i$ is achieved by mean-pooling over learned representations of nodes in $\mathcal{V}_i^D$, learned by the same GNN enocder in Eq. (4). Then we could achieve the invariance loss for optimizing the term $\mathbb{E}[\log P(Z|D, \widehat{G}_c)]$ in Theorem 3.1 as follows:

$$\mathcal{L}_d = \frac{1}{|\mathcal{V}|} \sum_{i=1}^{|\mathcal{V}|} \|\mathbf{h}_i - \mathbf{z}_i\|_2, \text{ where } \mathbf{z}_i \sim \mathcal{N}(\mathbf{z}|\mu_i, \text{diag}(\sigma_i^2)). \tag{12}$$

Here $\mathbf{z}_i$ is obtained in the same way as in Eq. (4). As such, the invariance loss could be used to alleviate the domain influence on learned $Z$, which may involve spurious information.

**Optimization.** With our derived losses, the overall GRM objective for optimization is formulated as follows:

$$\mathcal{L} = \mathbb{E}_{(G,Y) \sim \mathbb{P}(G,Y|D)} \mathbb{E}_{D \in \mathcal{D}_{tr}} \left[ \mathcal{L}_s(G, Y) + \alpha \mathcal{L}_r(G) + \beta \mathcal{L}_d(G, D) \right], \tag{13}$$

where $\alpha$ and $\beta$ are two hyper-parameters to control the weight of $\mathcal{L}_r$ and $\mathcal{L}_d$, respectively. In this way, we can effectively optimize our proposed GRM objective to tackle the OOD generalization on graph data.

### 3.4 Complexity Analysis

In this subsection, we analyze the time complexity of our framework. Particularly, the time complexity of our framework is primarily determined by the GNN encoder and the VGAE generator module, along with the three losses. Therefore, we first break down the complexity by considering the GNN and VGAE separately, then combining their contributions. Note that the time complexity of the GNN encoder is $O(|\mathcal{V}|d^2 + |\mathcal{E}|d)$. For the VGAE complexity, the module (1) encodes each node's representation and (2) reconstructs the node's representation. For each node, the VAE in VGAE performs operations involving encoding and decoding, which typically, and thus the time complexity for each node's VAE operation is proportional to $d^2$. Thus, for all nodes, the VAE complexity is $O(|\mathcal{V}|d^2)$. Note that this process already involves the time complexity of the regularization loss. For the remaining two losses, the supervision loss and the invariance loss, we compute the time complexity as follows. First, since we are using the cross-entropy loss as the supervision loss, the time complexity is $O(|\mathcal{V}|d)$. The invariance loss involves computing the Euclidean distance between

Table 1: Statistics of six out-of-distribution node classification datasets.

| Shift Type | Dataset | # Nodes | # Edges | # Classes | # Domains | Metric |
|---|---|---|---|---|---|---|
| Artificial | Cora | 2,703 | 5,278 | 10 | 1/1/8 | Accuracy |
| Transformation | Photo | 7,650 | 119,081 | 10 | 1/1/8 | Accuracy |
| Cross-Domain | Twitch | 1,912 - 9,498 | 31,299 - 153,138 | 2 | 1/1/5 | ROC-AUC |
| Transfers | FB-100 | 769 - 41,536 | 16,656 - 1,590,655 | 2 | 3/2/3 | Accuracy |
| Temporal | Elliptic | 203,769 | 234,355 | 2 | 5/5/9 | F1 Score |
| Evolution | Arxiv | 169,343 | 1,166,243 | 40 | 1/1/3 | Accuracy |

Table 2: The graph OOD generalization results (test accuracy in % for Cora, Photo, FB-100, and ROC-AUC in % for Twitch). The best results are in **bold**.

| Dataset | Cora | | Photo | | FB-100 | | Twitch | |
|---|---|---|---|---|---|---|---|---|
| Method | Min. | Avg. | Min. | Avg. | Min. | Avg. | Min. | Avg. |
| ERM | $65.0_{\pm1.5}$ | $68.2_{\pm0.4}$ | $84.4_{\pm1.5}$ | $88.6_{\pm1.3}$ | $50.5_{\pm0.4}$ | $52.8_{\pm0.6}$ | $49.7_{\pm1.1}$ | $52.2_{\pm0.9}$ |
| DRNN | $56.4_{\pm1.4}$ | $74.8_{\pm1.2}$ | $76.7_{\pm1.5}$ | $77.1_{\pm1.2}$ | $48.0_{\pm1.0}$ | $51.4_{\pm0.7}$ | $44.0_{\pm0.5}$ | $48.1_{\pm1.4}$ |
| MMD | $52.4_{\pm1.5}$ | $75.8_{\pm0.6}$ | $82.1_{\pm1.1}$ | $84.8_{\pm0.6}$ | $51.4_{\pm0.9}$ | $53.3_{\pm0.7}$ | $42.8_{\pm0.6}$ | $49.1_{\pm0.9}$ |
| ARM | $60.6_{\pm1.1}$ | $62.9_{\pm1.4}$ | $58.3_{\pm1.1}$ | $74.6_{\pm0.7}$ | $50.7_{\pm1.3}$ | $54.5_{\pm0.9}$ | $43.2_{\pm1.5}$ | $48.5_{\pm1.3}$ |
| EERM | $68.0_{\pm0.6}$ | $70.5_{\pm1.0}$ | $90.8_{\pm0.5}$ | $91.8_{\pm0.9}$ | $50.9_{\pm0.4}$ | $54.3_{\pm1.4}$ | $51.6_{\pm0.8}$ | $54.1_{\pm0.9}$ |
| LiSA | $71.1_{\pm1.5}$ | $76.7_{\pm0.8}$ | $90.3_{\pm1.2}$ | $91.5_{\pm1.5}$ | $48.8_{\pm1.2}$ | $54.2_{\pm1.0}$ | $48.6_{\pm1.2}$ | $55.8_{\pm2.2}$ |
| IS-GIB | $71.3_{\pm1.9}$ | $78.6_{\pm1.5}$ | $87.2_{\pm0.6}$ | $90.2_{\pm0.9}$ | $49.6_{\pm1.6}$ | $54.6_{\pm1.2}$ | $51.2_{\pm1.9}$ | $56.0_{\pm1.2}$ |
| MARIO | $70.8_{\pm1.3}$ | $76.1_{\pm1.0}$ | $88.6_{\pm0.8}$ | $89.4_{\pm1.4}$ | $50.3_{\pm1.9}$ | $53.9_{\pm1.4}$ | $50.7_{\pm2.0}$ | $55.1_{\pm1.9}$ |
| GRM | $\mathbf{74.2}_{\pm1.2}$ | $\mathbf{81.2}_{\pm1.5}$ | $\mathbf{91.3}_{\pm0.9}$ | $\mathbf{92.7}_{\pm1.6}$ | $\mathbf{52.0}_{\pm1.3}$ | $\mathbf{55.1}_{\pm1.1}$ | $\mathbf{52.5}_{\pm1.7}$ | $\mathbf{56.7}_{\pm1.0}$ |

two embeddings of each node, averaging across the graph. Therefore, the time complexity is $O(|\mathcal{V}|^2 d)$. In conclusion, the overall time complexity is calculated as

$$O(|\mathcal{V}|d^2 + |\mathcal{E}|d + |\mathcal{V}|d^2 + |\mathcal{V}|d + |\mathcal{V}|^2 d). \tag{14}$$

By simplifying the above time complexity, we can obtain the final time complexity as

$$O(|\mathcal{V}|d^2 + |\mathcal{E}|d + |\mathcal{V}|^2 d). \tag{15}$$

## 4 Experiments

### 4.1 Experimental Setup

**Datasets.** In our node-level OOD generalization experiments, we evaluate GRM and other state-of-the-art baselines on six real-world datasets that cover different topics and tasks, following EERM (Wu et al., 2022a). We summarize the statistics of these datasets in Table 1. Specifically, we use datasets that involve three different types of distribution shifts: (1) "*Artificial Transformation*" denotes that synthetic spurious features are added to these datasets; (2) "*Cross-Domain Transfers*" means that each domain in the datasets corresponds to a graph distinct from each other; (3) "*Temporal Evolution*" means that the datasets are dynamic with evolving nature. Each type includes two datasets. More details about these datasets can be found in Appendix B.

**Baselines.** We evaluate our GRM framework in comparison to two sets of baselines. The general OOD generalization methods include ERM, DRNN (Koh et al., 2021), MMD (Li et al., 2018), and ARM (Zhang et al., 2021). The state-of-the-art graph OOD generalization methods include EERM (Wu et al., 2022a), IS-GIB (Yang et al., 2023), MARIO (Zhu et al., 2023), and LiSA (Yu et al., 2023). We provide more details and the parameter settings of these baselines in Appendix C.3.

Table 3: The graph OOD generalization results (test accuracy in % for Arxiv and F1 score in % for Elliptic). The best results are in **bold**.

| Dataset | Elliptic | | | | Arxiv | | | |
|---|---|---|---|---|---|---|---|---|
| Method | T1 | T2 | T3 | Avg. | T1 | T2 | T3 | Avg. |
| ERM | $59.6_{\pm1.4}$ | $63.5_{\pm1.3}$ | $61.7_{\pm0.6}$ | $61.6_{\pm1.1}$ | $47.6_{\pm0.9}$ | $45.5_{\pm1.4}$ | $41.4_{\pm1.0}$ | $44.8_{\pm1.4}$ |
| DRNN | $73.2_{\pm1.4}$ | $71.4_{\pm0.7}$ | $70.6_{\pm0.3}$ | $71.8_{\pm0.8}$ | $46.8_{\pm0.5}$ | $44.7_{\pm1.1}$ | $40.5_{\pm1.3}$ | $44.0_{\pm1.0}$ |
| MMD | $71.9_{\pm0.7}$ | $70.1_{\pm0.4}$ | $69.9_{\pm0.8}$ | $70.6_{\pm0.8}$ | $44.6_{\pm1.3}$ | $42.4_{\pm0.7}$ | $38.9_{\pm1.0}$ | $42.0_{\pm0.5}$ |
| ARM | $72.1_{\pm1.5}$ | $69.7_{\pm0.7}$ | $67.9_{\pm1.4}$ | $69.9_{\pm1.3}$ | $44.9_{\pm0.7}$ | $42.3_{\pm0.6}$ | $39.7_{\pm0.8}$ | $42.3_{\pm1.0}$ |
| EERM | $66.3_{\pm0.4}$ | $63.8_{\pm0.6}$ | $55.5_{\pm0.6}$ | $61.9_{\pm1.1}$ | $50.3_{\pm1.4}$ | $48.3_{\pm0.4}$ | $44.7_{\pm1.4}$ | $47.8_{\pm1.4}$ |
| LiSA | $68.8_{\pm0.9}$ | $65.6_{\pm0.7}$ | $69.3_{\pm1.0}$ | $67.9_{\pm0.8}$ | $45.9_{\pm0.6}$ | $42.3_{\pm0.5}$ | $46.1_{\pm0.8}$ | $44.7_{\pm0.6}$ |
| IS-GIB | $71.2_{\pm1.1}$ | $70.0_{\pm1.0}$ | $70.4_{\pm1.2}$ | $70.5_{\pm1.1}$ | $49.3_{\pm0.8}$ | $46.6_{\pm0.9}$ | $50.5_{\pm1.3}$ | $48.8_{\pm0.7}$ |
| MARIO | $69.8_{\pm1.9}$ | $72.8_{\pm2.4}$ | $71.1_{\pm1.4}$ | $71.2_{\pm2.0}$ | $48.8_{\pm2.3}$ | $50.1_{\pm2.4}$ | $49.2_{\pm2.4}$ | $49.4_{\pm2.8}$ |
| GRM | $\mathbf{89.4}_{\pm1.5}$ | $\mathbf{85.5}_{\pm1.1}$ | $\mathbf{89.1}_{\pm1.5}$ | $\mathbf{88.0}_{\pm1.4}$ | $\mathbf{52.2}_{\pm0.9}$ | $\mathbf{52.6}_{\pm1.4}$ | $\mathbf{56.1}_{\pm1.4}$ | $\mathbf{53.6}_{\pm1.2}$ |

**GRM Settings.** In this subsection, we introduce the detailed parameter settings in our framework GRM. Specifically, we use the Adam optimizer (Kingma & Ba, 2015) for training. The dropout rate is set as 0.3, and the weight decay rate is 0.001. The learning rate is set as 0.01. Given an input graph, we utilize two 2-layer GCNs (Kipf & Welling, 2017), with a hidden dimension size of 128, to learn domain-specific representations and node representations. Then we concatenate these two representations as the input of our VAE-based generator. The encoder of the generator is also implemented as a 2-layer GCN. The dimension of latent variables (i.e., $d_z$) is set as 128. For the specific values of $L^*$ and $P^*$ in selecting nodes for learning domain-specific representations, we set them as 3 and 1.5, respectively. For the neighborhood size of the computation graph $G$ of node $v$, i.e., $L$, we set it as 2. In other words, two-hop neighbors will be included in the computation graph $G$. We run 5 times for this process and aggregate the classification results. We provide our code in the supplementary materials. During training, we conduct all experiments on one NVIDIA A6000 GPU with 48GB of memory. We adopt the same GNN encoder for all baselines, i.e., a 2-layer GCN (Kipf & Welling, 2017). Notably, since DRNN and MMD are not designed for scenarios with only one training domain, we use the interpolated domains generated by EERM as training domains for these two methods.

## 4.2 Comparative Results on Node-Level Tasks

To comprehensively demonstrate the effectiveness of GRM, we evaluate its performance on six node-level datasets with different types of distribution shifts and provide results in Table 2 and Table 3. We report the worst case result (Min.) and average result (Avg.) for the first four datasets since they consist of multiple (larger than three) test domains. The detailed results on each test domain are provided in Appendix D. From the results, we summarize the observations as follows:

- Across all datasets with various types of distribution shifts, GRM consistently outperforms all other baselines on both the worst case (Min.) and average (Avg.) results, which validates the superiority of GRM on graph OOD generalization of node classification tasks.

- The performance improvement of GRM over other baselines is substantially larger on Elliptic. This is because it contains a large number of test domains, leading to difficulties in generalizing to various test domains. Nevertheless, our GRM framework optimized with the invariance loss will provide better performance in this situation.

- The performance variances of GRM across test domains are lower on Photo and Arxiv compared to other baselines. These datasets preserve greater node degrees (i.e., more complex structures) and a larger class set. Our generative framework can learn more precise invariant subgraphs with the designed regularization loss.

### 4.3 Effect under Distribution Shifts

In this subsection, we evaluate the effectiveness of GRM under various degrees of distribution shift on the Cora dataset. We introduce artificial distribution shifts on Cora by mixing node features generated from labels and domain IDs (details are provided in Appendix E), and we refer to the modified dataset as Cora-Mix. We systematically evaluate our framework and other baselines on Cora-Mix under different spurious feature ratios and present the results in Fig. 3. The results show that the performance of all methods drops significantly when the bias ratio increases. The performance drop is particularly sharp when the bias ratio increases from 0 to 0.1, indicating that spurious features can adversely

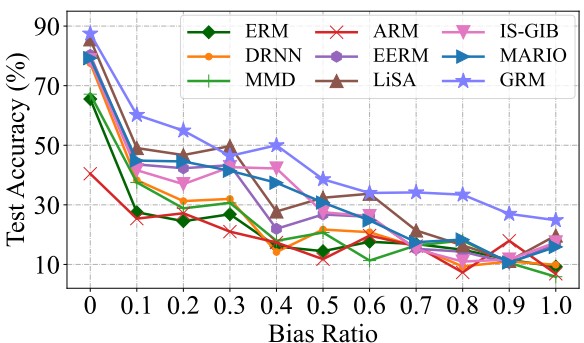

Figure 3: The results of various methods on dataset Cora-Mix with different degrees of distribution shifts.

affect all models even with a small ratio. Moreover, our proposed framework GRM consistently outperforms other baselines, especially when the bias ratio is relatively large (e.g., $0.5 \sim 0.9$). This demonstrates that GRM can effectively alleviate the adverse impact of spurious information by generating invariant subgraphs for classification.

### 4.4 Ablation Study

In this subsection, we perform a series of ablations studies to evaluate the efficacy of different components in our framework GRM. Specifically, we compare our proposed framework GRM with three degenerate versions: (1) GRM without the regularization loss, denoted as GRM\R; (2) GRM without the invariance loss. We denote this variant as GRM\I; (3) GRM without the VGAE-based generator, which means we remove the stochastic sampling of $Z$ during generation, denoted as GRM\V. From the results presented in Fig. 4, we obtain following insights. (1) GRM consistently outperforms its variants with different components removed, indicating that each module in GRM plays a vital role in handling distribution shifts. (2) Deprecating the invariance loss greatly reduces the performance on Twitch and FB-100 with a limited number of do-

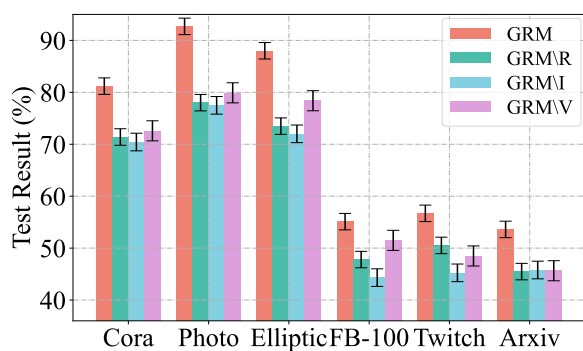

Figure 4: Ablation study of our framework GRM with different variants evaluated on six real-world datasets.

mains. This result implies that the invariance loss is crucial for datasets with few domains for existing works to learn invariant representations. (3) The performance decreases differently by removing the VGAE module or the regularization loss. Specifically, removing the regularization loss typically leads to a more significant performance drop, as it is more challenging to generate precise invariant subgraphs without regularization. Namely, the potential risk of overfitting is detrimental to performance. (4) From a broader perspective, the invariance loss generally plays a more critical role than the other two components, as its removal causes a larger performance drop. This is because learning invariant subgraphs is essential for addressing distribution shifts, as it directly impacts the effectiveness of a classifier trained on a domain different from the test domain. (5) Moreover, the regularization loss and the VGAE module contribute in complementary ways. The regularization loss prevents the generated graphs from deviating from a specific distribution, while the VGAE module introduces randomness during generation. Both are crucial for maintaining the diversity and robustness of the generated subgraphs. In summary, these components work together to enable GRM to achieve robust generalization across diverse datasets and distribution shifts.

## 4.5 Comparative Results on Graph-Level Tasks

Although we focus on the node classification task, our method is also applicable to graph classification, i.e., graph-level out-of-distribution generalization. In the setting for graph-level tasks, the domain information exists in other graphs and thus could not directly calculate the node influence. Thus, we still use the nodes in the input graph $G$ to learn domain-specific representations $H$. Notably, as these nodes will not cover the entire graph $G$, the learned $H$ will not be trivial, i.e., the same for all nodes in $G$. For graph-level experiments, We consider four prevalent datasets, namely **SP-Motif** (Ying et al., 2019), **MNIST-75sp** (Knyazev et al., 2019), **G-SST2** (Graph-SST2) (Socher et al.,

Table 4: The OOD graph classification results (ROC-AUC for Molhiv and accuracy in % for other datasets) of various methods on four datasets, with the best results in **bold**.

| Method | SP-Motif | MNIST | G-SST2 | Molhiv |
|--------|----------|-------|--------|--------|
| DIR    | 39.87    | 20.36 | 83.29  | 77.05  |
| GIL    | 46.04    | 21.94 | 83.44  | 79.08  |
| CIGA   | 64.01    | 25.29 | 81.02  | 79.75  |
| GALA   | 64.54    | 26.09 | 83.79  | 80.53  |
| GRM    | **65.05**| **26.53** | **83.86** | **81.02** |

2013), and **Molhiv** (OGBG-Molhiv) (Hu et al., 2020), with detailed provided in Appendix B.4. For baselines, we consider four state-of-the-art methods: DIR (Wu et al., 2022b), GIL (Li et al., 2022b), CIGA (Chen et al., 2022), and GALA (Chen et al., 2023). From the results presented in Table 4, we observe that GRM still exhibits competitive performance on OOD graph classification. Specifically, it achieves the best results over other baselines on all four datasets. The performance improvement is better on the dataset Molhiv with a larger graph size, thereby providing richer domain knowledge for our GRM to learn invariant information.

# 5 Related Works

## 5.1 Out-of-Distribution (OOD) Generalization

OOD Generalization aims to learn a model that can generalize to an unseen test domain, given several different but related training domain(s). Prior invariant methods (Ganin & Lempitsky, 2015; Li et al., 2018; Arjovsky et al., 2019) genreally focus on learning invariant features (Sun et al., 2016; Peng et al., 2019) or optimizing for the worst-case group performance (Hu et al., 2018; Sagawa et al., 2020). Recent works for OOD generalization on graphs (Chen et al., 2022; Li et al., 2022b; Wang et al., 2024) could be typically categorized into two classes: invariant learning and graph augmentation (Li et al., 2022a). Among invariant learning methods, CIGA (Chen et al., 2022) proposes to extract subgraphs that maximally preserve the invariant intra-class information based on causality. DIR (Wu et al., 2022b) uses a set of graph representations as the invariant rationales to create additional distributions. GIL (Li et al., 2022b) identifies invariant subgraphs via a GNN-based generator. More recently, MARIO (Zhu et al., 2023) utilizes the Information Bottleneck (IB) principle to learn invariant information. Among augmentation methods, LiSA (Yu et al., 2023) proposes to leverage graph augmentation to obtain more diverse training data for learning invariant information. EERM (Wu et al., 2022a) generates domains by maximizing the loss variance between domains in an adversarial manner, such that the obtained domains could aid in learning invariant representations.

## 5.2 Graph Generative Models

In recent years, numerous works have been proposed for graph generation (You et al., 2018; Grover et al., 2019). Specifically, GraphVAE (Simonovsky & Komodakis, 2018) proposes a framework based on VAE (Kingma & Welling, 2013) to generate graphs by encoding existing graphs. GraphRNN (You et al., 2018) generates graphs through a sequence of node and edge formations. Moreover, several methods (Jin et al., 2018; Preuer et al., 2018) focus on generating graphs based on specific knowledge. For example, MolGAN (De Cao & Kipf, 2018) adapts the framework of Generative Adversarial Networks (GANs) (Goodfellow et al., 2014) to operate directly on graph-structured data with a reinforcement learning objective. Note that although these methods leverage different information for generating graphs, they are not explicitly proposed for handling the distribution shift problem on graphs. In contrast, our framework GRM aims to utilize domain information to generate graphs that are suitable for a trained classifier.

# 6 Conclusion

In this paper, we propose a novel framework, namely Generative Risk Minimization (GRM), to generate invariant subgraphs for each input graph to tackle the OOD generalization problem on graphs. Instead of extracting structures that may cause the loss of invariant information, we propose our GRM objective that incorporates a generation term and a mutual information term. We derive three types of losses to enable the optimization of our GRM objective in the absence of ground truths for the invariant subgraphs. The effectiveness of GRM is validated by our theoretical analysis and also the extensive experiments across both node-level and graph-level OOD generalization tasks. The results indicate the superiority of GRM over other state-of-the-art baselines.

# Acknowledgments

This work is supported in part by the National Science Foundation under grants (IIS-2006844, IIS-2144209, IIS-2223769, CNS-2154962, BCS-2228534, and CMMI2411248), the Commonwealth Cyber Initiative Awards under grants (VV-1Q24-011, VV-1Q25-004), and the research gift funding from Netflix and Snap.

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

## A Theoretical Analysis

### A.1 Theorem 3.1 and Proof

In this section, we provide proof for Theorem 3.1.

**Theorem 3.1.** *An evidence lower bound (ELBO) for optimization of the GRM objective, by introducing a latent causal variable $Z$ and a variational approximation $Q(\widehat{G}_c)$, is as follows:*

$$\max \ \mathbb{E}\left[\log P(\widehat{G}_c|G,Z)\right] - KL(Q(Z|G)\|P(Z|G))$$
$$- \mathbb{E}[KL(P(\widehat{G}_c|D,Z)\|Q(\widehat{G}_c))] + \mathbb{E}\left[\log P(Z|D,\widehat{G}_c)\right], \tag{16}$$

*Proof.* We first present the GRM objective:

$$\max \ \mathbb{E}\left[\log P(\widehat{G}_c|G)\right] - I(\widehat{G}_c;D). \tag{17}$$

We first derive the ELBO for the generation objective, which is a standard derivation for the variational auto-enocder (VAE):

$$\log P(\widehat{G}_c|G)$$
$$= \log \int_Z P(\widehat{G}_c, Z|G)dZ$$
$$= \log \int_Z Q(Z|G)\frac{P(\widehat{G}_c, Z|G)}{Q(Z|G)}dZ$$
$$(\textit{using Jensen's Inequality})$$
$$\geq \int_Z Q(Z|G) \log \frac{P(\widehat{G}_c, Z|G)}{Q(Z|G)}dZ$$
$$= \mathbb{E}_Q[\log \frac{P(\widehat{G}_c, Z|G)}{Q(Z|G)}] \tag{18}$$
$$(\textit{using the property of conditional probabilities})$$
$$= \mathbb{E}_Q[\log \frac{P(\widehat{G}_c|Z, G) \cdot P(Z|G)}{Q(Z|G)}]$$
$$= \mathbb{E}_Q[\log P(\widehat{G}_c|Z, G)] - \mathbb{E}_Q[\log \frac{Q(Z|G)}{P(Z|G)}]$$
$$(\textit{using the definition of KL-divergence})$$
$$= \mathbb{E}_Q[\log P(\widehat{G}_c|G, Z)] - \mathrm{KL}(Q(Z|G)\|P(Z|G)).$$

Then we decompose the second term $-I(\widehat{G}_c;D)$ of our GRM objective as follows, based on the definition of mutual information:

$$-I(\widehat{G}_c;D) = I(\widehat{G}_c;Z|D) - I(\widehat{G}_c;D,Z) \tag{19}$$

We first decompose the first term:

$$I(\widehat{G}_c;Z|D) = \mathbb{E}\left[\log \frac{P(\widehat{G}_c|D,Z)}{P(\widehat{G}_c|D)}\right]$$
$$= \mathbb{E}\left[\log \frac{P(Z|D,\widehat{G}_c)}{P(Z|D)}\right] \tag{20}$$
$$= \mathbb{E}\left[\log P(Z|D,\widehat{G}_c)\right] + H(Z|D)$$

We consider $P(Z|D)$ as a deterministic distribution for each $D$, and thus it could be ignored for optimization. Then we dereive the lower bound for the second term:

$$
\begin{aligned}
-I(\widehat{G}_c; D, Z) &= -\mathbb{E}_{\widehat{G}_c, D}\left[\log\left(\frac{P(\widehat{G}_c|D, Z)}{P(\widehat{G}_c)}\right)\right] \\
&= -\mathbb{E}_{\widehat{G}_c, D}\left[\log\left(\frac{P(\widehat{G}_c|D, Z)}{Q(\widehat{G}_c)}\right)\right] + \text{KL}(P(\widehat{G}_c)\|Q(\widehat{G}_c)) \\
&\geq -\mathbb{E}_{\widehat{G}_c, D}\left[\log\left(\frac{P(\widehat{G}_c|D, Z)}{Q(\widehat{G}_c)}\right)\right] \\
&= -\mathbb{E}_G[\text{KL}(P(\widehat{G}_c|D, Z)\|Q(\widehat{G}_c))]
\end{aligned}
\tag{21}
$$

Finally, we could combine the above three derivation results to achieve the final evident lower bound for optimization of the GRM objective:

$$
\begin{aligned}
\max \ \ &\mathbb{E}\left[\log P(\widehat{G}_c|G, Z)\right] - \text{KL}(Q(Z|G)\|P(Z|G)) \\
&- \mathbb{E}[\text{KL}(P(\widehat{G}_c|D, Z)\|Q(\widehat{G}_c))] + \mathbb{E}\left[\log P(Z|D, \widehat{G}_c)\right],
\end{aligned}
\tag{22}
$$

$\square$

# B  Datasets in Experiments

In this section, we provide further details on the six datasets used in our experiments: `Cora`, `Photo`, `Twitch`, `FB-100`, `Elliptic`, and `Arxiv`. Note that the datasets are originally processed by EERM (Wu et al., 2022a). We follow the same dataset setting and splitting to keep consistency. Additionally, in Sec. 4.2, we report the Min. and Avg. performance on four datasets, and further detailed results of these datasets are presented in Appendix D.

## B.1  Artificial Distribution Shifts on `Cora` and `Photo`

`Cora` and `Photo` are two popular benchmark datasets used for node classification tasks and are also widely adopted to assess the effectiveness of GNN models. Specifically, these datasets are of moderate size, containing thousands of nodes (2,703 and 7,650, respectively). The data statistics are provided in Table 1. In particular, `Cora` is a citation network, where nodes represent papers and edges indicate the citation relationship between them. On the other hand, `Photo` is a co-purchasing network, with nodes representing specific goods and edges denoting frequent co-purchases of two goods. In the original dataset, the provided node features exhibit a strong correlation with node labels. Following EERM (Hamilton et al., 2017), in order to assess the model performance for graph OOD generalization under various distributions, we manually introduce distribution shifts into the training and testing data.

More specifically, we construct node labels and spurious domain-sensitive attributes from node features. Given the node features as $X_1$, we start by randomly initializing a Graph Neural Network (GNN) with $X_1$ as input and an adjacency matrix to generate node labels $Y$. To obtain the one-hot label vectors, we perform an argmax operation in the output layer. Then, we randomly initialize another GNN with the concatenation of $Y$ and a domain ID as input to generate spurious node features $X_2$. The next step is to concatenate these two sets of node features, i.e., $X = [X1, X2]$, to create new node features for training and test data. This process is performed ten times for each dataset, resulting in ten graphs with different domain IDs. For training and validation, we utilize one graph each, while the classification accuracy is reported on the remaining graphs.

### B.2 Cross-Domain Transfers on `Twitch` and `FB-100`

Cross-domain transfers are a common occurrence in scenarios involving distribution shifts on graphs. In various real-world situations, multiple observed graphs are available, each belonging to a specific domain. For instance, in social networks, domains can be defined based on where or when the networks are collected. Similarly, in protein networks, distinct species may have their own observed graph data, such as protein-protein interactions, representing separate domains. The key point is that graph data typically captures relational structures among specific entities, which often exhibit unique characteristics for different entity groups. As a result, the data-generating distributions can vary across these groups, leading to domain shifts.

However, in order to facilitate transfer learning across graphs, it is necessary for the graphs within a dataset to share the same input feature space and output space. To achieve this requirement, we utilize two publicly available datasets, `Twitch` and `FB-100`, which satisfy these conditions.

Specifically, the `Twitch` dataset consists of seven networks, where nodes and edges represent Twitch users and their mutual friendships, respectively. These networks are collected from different regions, namely DE, ENGB, ES, FR, PTBR, RU, and TW. Although these networks have similar sizes, they exhibit variations in terms of density and maximum node degrees, as presented in Table 5.

The `FB-100` dataset comprises 100 snapshots of Facebook friendship networks from 2005. Here each network contains nodes that represent Facebook users from a specific American university. In our experiments, we utilize fourteen networks: John Hopkins, Caltech, Amherst, Bingham, Duke, Princeton, WashU, Brandeis, Carnegie, Cornell, Yale, Penn, Brown, and Texas. These graphs exhibit significant variations in terms of sizes, densities, and degree distributions, indicating that the model capability in handling different graph structures becomes crucial for this dataset.

### B.3 Temporal Evolution on Dynamic Graph Data: `Elliptic` and `Arxiv`

The distribution shift problem can also occur in temporal graphs that dynamically change over time. The evolution of these graphs can be generally categorized into two types. In the first type, there exist multiple snapshots of the graph, with each snapshot captured at a specific time. As time progresses, a sequence of graph snapshots is generated, which may exhibit variations in terms of node sets and data distributions. For example, financial networks capture the payment flows among transactions within different time intervals and thus result in different domains. In the second type, there exists only one single graph that evolves through the addition or deletion of nodes and edges. This type is commonly seen in large-scale real-world graphs, such as social networks and citation networks. In these graphs, the distribution of node features, edges, and labels often exhibit a strong correlation with time at different scales. For our graph OOD generalization experiments, we utilize two public real-world datasets, namely `Elliptic` and `Arxiv`. These datasets are suitable for exploring node classification tasks within the context of evolving temporal graphs.

The `Elliptic` dataset consists of a series of 49 graph snapshots, where each snapshot represents a network of Bitcoin transactions. Specifically, each node corresponds to a transaction and each edge represents a payment flow. Within these transactions, approximately 20% are labeled as either licit or illicit, with the objective being to identify illicit transactions within future networks. In the original dataset, the first six graph snapshots contain highly imbalanced classes, with the number of illicit transactions being less than 10 among thousands of nodes. Consequently, we exclude these snapshots and focus on the 7th to 11th, 12th to 17th, and 17th to 49th snapshots for training, validation, and testing, respectively. Furthermore, due to the low positive label rate observed in each graph snapshot, we organize the 33 testing graph snapshots into 9 distinct test sets based on their chronological order. The dataset also requires the framework to effectively handle diverse label distributions encountered during the transition from training to testing data.

The `Arxiv` dataset comprises 169,343 Arxiv CS papers covering 40 subject areas, along with their citation relationships. The objective is to predict the subject area of a given paper. In (Hu et al., 2020), the papers published before 2017, in 2018, and since 2019 were utilized for training, validation, and testing, respectively. They employed a transductive learning setting (Tan et al., 2022c; Dolz et al., 2020; Mathavan et al., 2023), wherein the nodes in the validation and test sets were also present in the training graph. Instead, `Arxiv` adopts an inductive learning setting, which better reflects real-world scenarios. Here, the nodes in the

validation and test sets are unseen during training, introducing a greater level of novelty. Specifically, the dataset consists of papers published before 2011 for training, papers from 2011 to 2014 for validation, and papers after 2014 for testing. Such a splitting strategy introduces a distribution shift between the training and testing data, as specific latent influential factors (such as research topic popularity) in data generation would change over time.

Table 5: Statistics of `Twitch` dataset.

| **Domain** | # Nodes | # Edges | Density | Avg. Degree | Max. Degree |
|:---:|:---:|:---:|:---:|:---:|:---:|
| DE | 9,498 | 153,138 | 0.0033 | 16 | 3,475 |
| ENGB | 7,126 | 35,324 | 0.0013 | 4 | 465 |
| ES | 4,648 | 59,382 | 0.0054 | 12 | 809 |
| FR | 6,549 | 112,666 | 0.0052 | 17 | 1,517 |
| PTBR | 1,912 | 31,299 | 0.0171 | 16 | 455 |
| RU | 4,385 | 37,304 | 0.0038 | 8 | 575 |
| TW | 2,772 | 63,462 | 0.0165 | 22 | 1,171 |

### B.4 Graph Classification Datasets for Out-of-Distirbution Generalization

Here we introduce the four datasets used in our experiments in Sec. 4.5, focusing on graph classification tasks. In particular, we leverage the following datasets:

- **SP-Motif** (Ying et al., 2019): The SP-Motif dataset, comprising 18,000 synthetic graphs, is constructed by combining a base graph (denoted by Tree, Ladder, or Wheel, represented as $S = 0, 1, 2$ respectively) with a motif (Cycle, House, or Crane, denoted as $C = 0, 1, 2$ respectively). The label $Y$ of each graph is exclusively determined by its motif $C$. In the construction of the training set, deliberate spurious correlations between the base $S$ and the label $Y$ are introduced. These correlations are quantified by the formula $P(S) = b \times \mathbb{I}(S = C) + (1 - b)/2 \times \mathbb{I}(S \neq C)$, where the motif follows a uniform distribution, and the base's distribution is contingent on the motif. The parameter $b$ is varied to produce different levels of bias within the Spurious-Motif datasets. The testing set features randomly combined motifs and bases, including graphs with larger bases, to accentuate the distributional disparities. In our experiments, we set the value of $b$ as 0.9.

- **MNIST-75sp** (Knyazev et al., 2019): This dataset transforms MNIST images into 70,000 graphs, each comprising up to 75 superpixels. These superpixels, which serve as graph nodes, are interconnected based on their spatial proximity, forming the graph edges. Each graph is categorized into one of 10 classes. Notably, the testing set is augmented with random noise in the node features to introduce variability.

- **Graph-SST2** (Socher et al., 2013; Yuan et al., 2022): This dataset includes graphs labeled according to sentence sentiment, with nodes representing tokens and edges reflecting the syntactic relationships between them. The graphs in this dataset are partitioned into different subsets based on their average node degree, thereby creating dataset shifts.

- **Molhiv** (OGBG-Molhiv) (Wu et al., 2018; Hu et al., 2020): Designed for the task of molecular property prediction, the Molhiv dataset encompasses graphs that represent molecules, where nodes correspond to atoms and edges denote chemical bonds. Each graph is labeled based on its efficacy in inhibiting HIV replication.

## C Implementation Details

### C.1 Baseline Settings

In this subsection, we introduce the detailed settings for baseline methods used in our experiments.

- **ERM**: ERM denotes Empirical Risk Minimization, which conducts learning across domains without designs for distribution shifts. This baseline acts as a vanilla comparison where no specific techniques are employed for OOD generalization. We use the same GNN encoder as our framework and set the learning rate as 0.001.

- **EERM** (Wu et al., 2022a): EERM denotes Explore-to-Extrapolate Risk Minimization, which minimizes the mean and variance of risks from multiple domains that are simulated by adversarial context generators. For the parameter setting, we follow the choice in the original paper and search the hyper-parameters with grid search on the validation dataset. Specifically, the learning rate for GNN is chosen from $\{0.0001, 0.0002, 0.001, 0.005, 0.01\}$, the learning rate for graph editors is from $\{0.0001, 0.001, 0.005, 0.01\}$, and the weight for combination is chosen from $\{0.2, 0.5, 1.0, 2.0, 3.0\}$.

- **ARM** (Zhang et al., 2021): ARM denotes Adaptive Risk Minimization, which adapts the model parameters to various domains based on a small batch of data from the domain. The ARM framework consists of three variants: ARM-CML, ARM-BN, and ARM-LL. In our experiments, we compare the variant of ARM-CML, which significantly outperforms all variants. Since ARM is not explicitly designed for graph data, we employ the identical GNN architecture to our framework as the encoder and randomly select nodes from each domain for adaptation. Following the original setting in the paper, we set the learning rate as 0.0001, the weight decay rate as 0.0001, and the support size as 50. Furthermore, for ARM-CML, the number of context channels is set as three.

- **DRNN** (Koh et al., 2021): DRNN aims to tackle the distribution shift problem by ensuring that the distribution minority receives sufficient training. Following the setting in ARM, we set the learning rate as 0.0001 and the robust step size as 0.01.

- **MMD** (Li et al., 2018): MMD aims to maximize the mean discrepancy across domains. For the parameter setting, we set the learning rate as 0.0001 and the gamma value as 1. The support size is set the same as ARM as 50.

- **IS-GIB** (Yang et al., 2023): IS-GIB aims to discard spurious features while learning invariant features from a high-order perspective via preserving class relationships under various distribution shifts. We follow the parameter setting in their code and set the learning rate as 0.01.

- **MARIO** (Zhu et al., 2023): MARIO proposes to simultaneously achieve generalizable representations while obtaining invariant representations via adversarial data augmentations, based on graph contrastive learning strategies (Oord et al., 2018; Khosla et al., 2020; Tan et al., 2022b; Wang et al., 2023). The learning rate is set as 0.001.

- **LiSA** (Yu et al., 2023): LiSA proposes to leverage variational subgraph generators to extract locally predictive patterns that could be used for constructing label-invariant subgraphs. These subgraphs are then used to create augmented environments with enhanced diversity.

We adopt the same GNN encoder for all baselines, i.e., a 2-layer GCN (Kipf & Welling, 2017). Since DRNN and MMD are not designed for scenarios with only one training domain, we use the interpolated domains generated by EERM as training domains for these two methods.

### C.2 Training Details

During training, we conduct all experiments on one NVIDIA A6000 GPU with 48GB of memory. The package requirements of our experiments are listed below.

- Python == 3.7.10
- torch == 1.8.1
- numpy == 1.18.5
- scipy == 1.5.3

- networkx == 2.5.1

- scikit-learn == 0.24.1

- pandas == 1.2.3

### C.3 Training Time and Memory Usage

In this subsection, we provide the time/memory of all experiments conducted on one NVIDIA A6000 GPU with 48GB of memory in Table 6.

Table 6: Training time and memory usage for different datasets.

| Dataset | Total Training Time (s) | Time Per Epoch (s) | Memory Usage (MB) |
|---------|-------------------------|--------------------|--------------------|
| Cora | 1,782.96 | 3.91 | 1,329 |
| Photo | 3,504.65 | 24.50 | 6,255 |
| Twitch | 1,156.37 | 7.93 | 3,011 |
| FB-100 | 5,902.70 | 44.05 | 30,387 |
| Elliptic | 539.43 | 16.19 | 4,103 |
| Arxiv | 4,355.21 | 4.50 | 5,505 |

From the results, we observe that all training times and memory usages are within a controllable range, demonstrating that our method is applicable to a wide variety of scenarios with different types of distribution shifts. Moreover, the training time per epoch is particularly higher for Elliptic, as graphs in this dataset contain significantly larger numbers of nodes and edges. Nevertheless, the total training time remains reasonable, indicating that our method is scalable to large datasets.

## D Detailed Results

In this section, we provide detailed results for the specific test domains on `Cora` (8 test domains), `Photo` (8 test domains), `FB-100` (3 test domains), and `Twitch` (5 test domains). The results are provided in Table 7, Table 8, and Table 9.

## E Created Datasets with Different Degrees of Distribution Shifts

In this section, we introduce the details of the dataset `Cora-Mix` used in Sec. 4.3. Specifically, we aim to manually control the degree of distribution shifts across different domains. However, the original dataset `Cora` provided in EERM (Wu et al., 2022a) creates distribution shifts via the concatenation of domain-sensitive features and label-related features, which means the degree of distribution shifts cannot be easily quantified. Therefore, we propose to mix up these two types of features with a weight to control the distribution shift degree. In particular, we follow the same strategy of generating these features, except that we changed their dimensions to be equal. In this manner, we can perform mix-up on them with a specific weight, i.e., the bias ratio. We also keep the domain split setting of 1/1/8 for training, validation, and test, respectively.

## F Limitations

Despite its superior performance, our framework still possesses several limitations. For example, our GRM framework involves the learning of domain information from other nodes or graphs in the domain. However, in practice, the available graphs in each domain may not be sufficient. As a result, the performance of our framework may be impacted. In addition, although our GRM framework is validated in both node-level and graph-level tasks, its performance on edge-level tasks is not evaluated.

Table 7: The graph OOD generalization results on `Cora`.

| Method | T1 | T2 | T3 | T4 | T5 | T6 | T7 | T8 |
|--------|-----|-----|-----|-----|-----|-----|-----|-----|
| ERM | 67.86 | 65.03 | 71.25 | 66.28 | 70.34 | 66.72 | 70.53 | 67.58 |
| DRNN | **76.62** | 73.63 | 83.09 | 56.35 | 78.18 | 78.47 | 79.84 | 72.49 |
| MMD | 75.30 | **86.23** | 82.87 | 52.44 | **82.02** | 77.70 | 80.61 | 69.05 |
| ARM | 62.22 | 64.25 | 62.74 | 62.27 | 64.11 | 62.92 | 60.64 | 64.40 |
| EERM | 70.09 | 70.99 | 72.55 | 71.13 | 71.03 | 68.04 | 71.29 | 68.88 |
| LiSA | 76.05 | 74.39 | 81.26 | 71.08 | 78.56 | 79.98 | 81.27 | 71.15 |
| IS-GIB | 75.67 | 76.41 | 84.61 | 74.07 | 82.90 | 80.79 | 82.74 | 71.32 |
| MARIO | 73.50 | 71.01 | 80.59 | 70.84 | 78.68 | 80.90 | 78.94 | **74.42** |
| GRM | 74.92 | 84.95 | **89.15** | **76.83** | 79.74 | **85.04** | **84.63** | 74.23 |

Table 8: The graph OOD generalization results on `Photo`.

| Method | T1 | T2 | T3 | T4 | T5 | T6 | T7 | T8 |
|--------|-----|-----|-----|-----|-----|-----|-----|-----|
| ERM | 84.35 | 89.57 | 89.39 | 90.14 | 87.63 | 90.08 | 87.34 | 90.04 |
| DRNN | 77.08 | 77.28 | 76.71 | 76.86 | 77.33 | 77.33 | 77.01 | 77.18 |
| MMD | 83.83 | 82.09 | 86.26 | 85.50 | 83.90 | 84.98 | 86.78 | 84.80 |
| ARM | 82.05 | 69.62 | 74.03 | 77.23 | 86.18 | 58.33 | 69.49 | 79.59 |
| EERM | 92.70 | 92.18 | **92.20** | 90.78 | 92.14 | 91.46 | 91.11 | 91.80 |
| LiSA | 92.01 | 92.14 | 90.88 | 90.26 | 91.62 | 91.09 | **92.24** | 91.71 |
| IS-GIB | 89.92 | 92.47 | 90.38 | 90.08 | 93.21 | 90.24 | 87.15 | 88.33 |
| MARIO | 91.18 | 89.24 | 89.37 | 88.34 | 89.81 | 88.87 | 88.57 | 90.10 |
| GRM | **93.37** | **92.53** | 91.91 | **93.86** | **94.33** | **91.56** | 91.30 | **92.42** |

Table 9: The graph OOD generalization results on `FB-100` and `Twitch`.

| Dataset | FB-100 | | | Twitch | | | | |
|---------|-----|-----|-----|-----|-----|-----|-----|-----|
| Method | T1 | T2 | T3 | T1 | T2 | T3 | T4 | T5 |
| ERM | 50.48 | 54.53 | 53.23 | 54.20 | 55.20 | 50.41 | 51.58 | 49.73 |
| DRNN | 49.56 | 56.76 | 47.98 | 50.92 | 53.33 | 43.98 | 45.87 | 46.47 |
| MMD | 51.35 | 57.00 | 51.58 | 53.94 | 54.13 | 42.81 | 48.30 | 46.30 |
| ARM | 50.73 | 56.64 | 56.11 | 52.18 | 49.49 | 43.24 | 50.21 | 47.13 |
| EERM | 50.85 | **56.73** | 55.39 | 57.19 | 55.17 | 51.61 | 52.54 | 53.79 |
| LiSA | **59.13** | 48.83 | 54.73 | **63.04** | **57.94** | 48.58 | 54.45 | 55.13 |
| IS-GIB | 49.55 | 53.25 | **60.87** | 61.23 | 57.08 | 51.69 | **54.50** | **55.36** |
| MARIO | 50.33 | 55.73 | 55.57 | 59.45 | 56.81 | 53.57 | 50.67 | 54.98 |
| GRM | 51.95 | 56.11 | 57.33 | 61.45 | 56.82 | **60.25** | 52.52 | 52.64 |

