# OpenReview forum: "Generative Risk Minimization for Out-of-Distribution Generalization on Graphs"
_TMLR — Accepted by TMLR_

### Review · Reviewer_E2za · 2024-11-21

**Summary Of Contributions:**

The paper proposes a 'Generative Risk Minimization' (GRM) framework to improve OOD generalization on graphs. Especially for node and graph classification tasks. For a given graph, the GRM framework generates an invariant sub-graph and uses it for node/graph classification. The loss function (objective) is a combination of a generation objective and an invariance objective, which does not rely on ground truth invariant graphs.

**Audience:**

Yes

**Claims And Evidence:**

No

**Requested Changes:**

**Required major changes**
1. In equation (4), please clarify what $\text{GNN}_{\mu}$ denotes. Is it the mean of node features after passing through the GNN?
2. Please add details about the GNN/VGAE architecture in the main paper as well. Right now, I could only find it in Appendix C The details about the optimizer used for training is also missing.
3. The complexity analysis in Section 3.4 does not seem to consider the depth of the GNN and VGAE. Additionally, there are no details regarding the choice of data/hidden state dimensions.
4. Please add a formal definition of $Q(Z)$. Right now, it is hard to understand what $Q(.)$ is. Also, in Section 3.2, the latent variable Z obtained via the VGAE shares the same notation as the actual Z which the authors aim to approximate. This notation inconsistency is confusing. See also the line above Eq (8).
5. Throughout the paper, the usage of the term 'continuous values of edges' is misleading. As per Eq (6), it seems like the invariant graph $\hat{G}_c$ is a complete graph and seems like the authors are referring to this structure as 'continuous values of edges'? Please clarify.
6. The details about the selection of influential nodes is a bit confusing. As per the Figure 2, it seems like the authors clearly distinguish between the nodes belonging to graph G and the influential nodes. However, no such demarcation was made in  Eq (10). Am I missing something? Also, how does this influential node selection vary for graph and node classification tasks? Please clarify.

**Strengths And Weaknesses:**

**Strengths** : The approach of leveraging causal variables to avoid the dependence on ground truth invariant graphs and its combination with the regularization loss seems interesting.

**Weaknesses:**

1. The presentation of the paper needs to be improved as it is currently unclear as to how the variational forms of latent variables are actually computed and used. My confusion arises from inconsistent notations.
2. Details about the computational aspects of the approach is lacking. In particular, details about the VGAE, the time/memory overheads of training with the GRM approach seem to be lacking.
3. In Appendix D, the GRM approach does not seem to do well on cross-domain OOD generalization with FB-100 and Twitch datasets. Can the authors explain this discrepancy when comparing the results in Table 2 and Table 7?

---

> ### Author Response · Authors · 2024-12-24
> **Response to Reviewer E2za**
>
> Dear Reviewer E2za,
>
> We sincerely appreciate your thoughtful feedback and constructive suggestions. Below, we address each of the weaknesses and requested changes raised in your review:
>
>
> ---
>
> ## **Weakness**
>
>
> ### **1. Presentation**
>
> > **Comment:** The presentation of the paper needs to be improved as it is currently unclear as to how the variational forms of latent variables are actually computed and used. My confusion arises from inconsistent notations.
>
> **Response**: Thank you for pointing this out. We have accordingly revised the content in Section 3.3 to provide further clarification. We also addressed your concerns regarding the notations and definitions in our following response to **Required Changes** 1, 4, 5, and 6.
>
>
> ---
>
>
> ### **2. Computational Aspects**
>
> > **Comment:** Details about the computational aspects of the approach is lacking. In particular, details about the VGAE, the time/memory overheads of training with the GRM approach seem to be lacking.
>
>
> **Response**: Thank you for the suggestion. We have provided the time/memory of all experiments conducted on an NVIDIA A6000 GPU with 48GB of memory. The results are as follows, and we have included them in Appendix C.3 in the revised version.
>
> | Dataset   | Total Training Time (s) | Time Per Epoch (s)| Memory Usage (MB) |
> |-----------|------------|-------|-------------------|
> | Cora      |  1,782.96   |3.91         |  1,329           |
> | Photo     |     3,504.65   |24.50       | 6,255               |
> | Twitch    | 1,156.37    |7.93          | 3,011              |
> | FB-100    |   5,902.7  |44.05        | 30,387       |
> | Elliptic  | 539.43  |16.19         | 4,103              |
> | Arxiv     |   4,355.21  |4.50       | 5,505             |
>
> From the results, we observe that all training times and memory usages are within a controllable range, demonstrating that our method is applicable to a wide variety of scenarios with different types of distribution shifts. Moreover, the training time per epoch is particularly higher for Elliptic, as graphs in this dataset contain significantly larger numbers of nodes and edges. Nevertheless, the total training time remains reasonable, indicating that our method is scalable to large datasets.
>
>
> ---
>
>
> ### **3. Cross-Domain OOD Generalization Performance**
>
> > **Comment:** In Appendix D, the GRM approach does not seem to do well on cross-domain OOD generalization with FB-100 and Twitch datasets. Can the authors explain this discrepancy when comparing the results in Table 2 and Table 7?
>
> **Response**: We would like to clarify that as stated in the first paragraph of Section 4.2: ``We report the worst case result (Min.) and average result (Avg.) for the first four datasets since they consist of multiple (larger than three) test domains.`` In other words, the results of FB-100 and Twitch presented in Table 2 are the min average values of multiple test domains, which are presented in Table 7. In fact, although our method GRM is not always the best on individual test domains of FB-100, the min values and average values (the more universal metric) **remain the best** across baselines.
>
>
>
> ---
>
> ## **Required Changes**
>
> ### **1. Clarification of Notations**
> > **Comment:** In equation (4), please clarify what $\text{GNN}_\mu$ denotes. Is it the mean of node features after passing through the GNN?
>
> **Response:** Here  GNN$\mu$  represents a GNN that processes the entire graph, and the resulting $\mu_{i}=\text{GNN}_\mu (\mathcal{V}, \mathcal{E}, X)_i$ denotes the $i$-th row of the output, i.e., the learned feature of the $i$-th node $v_i$. We have added an explanation to help clarify this in our revised version.
>
> ---
>
>
> ### **2. Setting Details**
>
> > **Comment:** Please add details about the GNN/VGAE architecture in the main paper as well. Right now, I could only find it in Appendix C The details about the optimizer used for training is also missing.
>
> **Response**: Thank you for the suggestion. We have moved the content in Appendix C to the main content in Section 4.1 and added the details for the optimizer.
>
>
> ---
>
>
> ### **3. Complexity Analysis Regarding Depth of GNN/VGAE**
>
> > **Comment:** The complexity analysis in Section 3.4 does not seem to consider the depth of the GNN and VGAE. Additionally, there are no details regarding the choice of data/hidden state dimensions.
>
> **Response**: Thank you for pointing out the relationship between our complexity analysis and the depth of the GNN and VGAE. We would like to clarify that an increase in the number of GNN/VGAE layers results in a linear increase in complexity, which is why it is omitted in the $O(\cdot)$ complexity analysis. Additionally, we have specified in our settings that we use a 2-layer GCN, which is a simple architecture and does not incur significant computational overhead. Moreover,

---

> ### Author Response · Authors · 2024-12-24
> **Response to Reviewer E2za (2/2)**
>
> ### **4. Formal Definition of $Q(Z)$**
>
> > **Comment:** Please add a formal definition of $Q(Z)$. Right now, it is hard to understand what $Q(\cdot)$ is. Also, in Section 3.2, the latent variable Z obtained via the VGAE shares the same notation as the actual Z which the authors aim to approximate. This notation inconsistency is confusing. See also the line above Eq (8).
>
> ---
>
> **Response**: Thank you for pointing this out. The latent variable $Z$ is introduced to derive the evidence lower bound (ELBO) as outlined in Theorem 3.1 of our paper. Additionally, we introduce the approximation $Q(Z)$, which represents a distribution of $Z$, allowing $Z$ to be learned through $Q(\cdot)$.
>
> Specifically, $Q(\cdot)$, used to model $Z$, is implemented as our generator (i.e., VGAE), as described in the first paragraph of Section 3.2. To ensure consistency, we use the same notation for $Z$ throughout, as it represents the variable we aim to learn. The VGAE implementation of $Q(\cdot)$ is explicitly designed to facilitate this learning process.
>
> We have updated the explanation in Section 3.2 in the revised version of the manuscript to provide further clarification.
>
>
> ---
>
>
> ### **5. Clarification of 'Continuous Values of Edges'**
>
>  > **Comment:** Throughout the paper, the usage of the term 'continuous values of edges' is misleading. As per Eq (6), it seems like the invariant graph $\widehat{G}_{c}$ is a complete graph and seems like the authors are referring to this structure as 'continuous values of edges'? Please clarify.
>
> **Response**: To clarify, we use the term 'continuous edge weights' to represent a value of $\hat{e}\_{ij}$ between 0 and 1, which is the weight of the edge between node $v_i$ and $v_j$. The value of $\hat{e}\_{ij}$ can be continuous, as we do not sample discrete edges (in which case, $\hat{e}\_{ij}$ is either 0 or 1). In this manner, the invariant graph $\widehat{G}\_{c}$ is a graph with continuous edge weights and could be processed by GNNs.
>
> ---
>
>
> ### **6. Selection of Influential Nodes**
>
>  > **Comment:** The details about the selection of influential nodes is a bit confusing. As per Figure 2, it seems like the authors clearly distinguish between the nodes belonging to graph G and the influential nodes. However, no such demarcation was made in Eq (10). Am I missing something? Also, how does this influential node selection vary for graph and node classification tasks? Please clarify.
>
> **Response**: As we state in just above Eq. (10): In practice, we choose the one-hop neighboring node set $\mathcal{N}_i$ of $v_i$ and select nodes that are most influential to $\mathcal{N}_i$ to maximally capture domain information. In Figure 2, the one-hop neighbor set $\mathcal{N}_i$ is represented as the input graph, along with the target node for classification. The influential nodes are decided by Eq. (10), which are the nodes with large node influence on $\mathcal{N}_i$. Lastly, the influential node selection is almost the same for graph and node classification tasks, except that the input graph for graph classification is an entire graph, while the input graph for node classification is the target node along with its 1-hop neighboring nodes.
>
>
> ---
>
> Thank you for your constructive feedback. We sincerely hope that our responses can address your concerns. Thanks again for your time and consideration.
>
> Best,
> The Authors

---

> ### Comment · Reviewer_E2za · 2024-12-28
>
> Thank you for clearing some of the concerns.
>
> The usage of the entire graph for graph classification tasks make sense. But, I am not convinced that only a 1-hop neighborhood of a node can be treated as a superset of influential nodes for node classification tasks. Although theoretical results in the literature convey that the influence decays exponentially with hops, there are usually constants involved in the bounds which can result in varying practical behavior as per the graph structures (homophily/heterophily/sparsity etc).

---

> > ### Author Response · Authors · 2024-12-30
> > **Reponse to Reviewer E2za**
> >
> > Dear Reviewer E2za,
> >
> > Thank you for your thoughtful feedback.
> >
> > We agree that multiple factors, such as homophily and heterophily, can influence node classification tasks. However, in our work, we **do not limit our approach to only 1-hop neighbors**. Specifically, the input graph only includes 1-hop neighbors, but we also select influential nodes to learn **domain-specific representations**, as described in Eq. (10). $\mathcal{N}_i$ represents the input graph node set and $\mathcal{V}_i^D$ denotes the selected node set for domain-specific representation, which can be potentially far away from the center node. Consequently, the information utilized for node classification extends beyond the immediate 1-hop neighborhood.
> >
> > We appreciate your insights and will further clarify this point in the revised version of our work.
> >
> > Best,
> > The Authors

---

### Review · Reviewer_EFEr · 2024-11-24

**Summary Of Contributions:**

- The authors propose Generative Risk Minimization (GRM), a framework that addresses OOD generalization on graphs. Unlike previous approaches that extract discrete invariant subgraphs, GRM generates invariant subgraphs with continuous edge weights while preserving the original node-set. The framework employs a Variational Graph Auto-Encoder (VGAE) architecture to generate invariant subgraphs. It is evaluated on six node-level and four graph-level real-world datasets covering various distribution shifts (artificial, cross-domain, temporal), consistently outperforming state-of-the-art baselines.

**Audience:**

Yes

**Claims And Evidence:**

Yes

**Requested Changes:**

Please see my concerns in the weaknesses section in my review.

[Typos]

On page 2, third paragraph:
- 'its' should be 'it'
- 'unaviable' should be 'unavailable'

**Strengths And Weaknesses:**

### Strength
- The proposed method is well-motivated.
- The paper provides rigorous theoretical analysis that effectively validates the tractable objective functions of the GRM framework.
- The performance of the proposed model is verified by both node-level and graph-level prediction tasks.

### Weakness
- While generating invariant subgraphs is an appealing idea, the paper lacks theoretical analysis for concept shift scenarios where $P(Y|X)$ changes between train and test distributions while $P(X)$ remains identical. The effectiveness of the proposed method under such concept shifts remains uncertain. The paper would be more solid if it either provides theoretical guarantees for concept shift scenarios or clearly specifies which type of distribution shift (covariate shift vs concept shift) each dataset exhibits.
- While the influential node extraction method appears reasonable for transductive learning scenarios, the paper lacks clear explanation for handling inductive learning cases like molecular property prediction, where only the input graph is available without access to other domain graphs. Although the authors mention using nodes within input graph $\mathcal{G}$ to learn domain-specific representations $H$ for graph-level tasks, exploring the use of virtual nodes could be an interesting direction as they might better capture domain-specific information without relying on the existing node set.

---

> ### Author Response · Authors · 2024-12-24
> **Response to Reviewer EFEr**
>
> Dear Reviewer EFEr,
>
> We sincerely thank you for your detailed feedback and valuable suggestions. Below, we would like to address each of the weaknesses raised:
>
> ---
>
> ### **1. Lack of Theoretical Analysis for Concept Shift Scenarios**
> >  **Comment:** While generating invariant subgraphs is an appealing idea, the paper lacks theoretical analysis for concept shift scenarios where $P(Y|X)$ changes between train and test distributions while  $P(X)$ remains identical. The effectiveness of the proposed method under such concept shifts remains uncertain. The paper would be more solid if it either provides theoretical guarantees for concept shift scenarios or clearly specifies which type of distribution shift (covariate shift vs concept shift) each dataset exhibits.
>
>
>
> **Response:** We acknowledge your concern regarding the lack of theoretical analysis for concept shift scenarios where $P(Y|X)$ changes between train and test distributions while $P(X)$ remains identical. We would like to clarify that our framework already addresses this scenario through the design of the **supervision loss** in our optimization process.
>
> Specifically, in Section 3.3, we propose a supervision loss based on the term $\mathbb{E}[\log P(\hat{G_c} \mid G, Z)]$, where $\hat{G_c}$ represents the generated subgraph. Since the ground-truth $G_c$ is unobserved, we use the label $Y$ of $G$ as a proxy for $G_c$, under the assumption that the optimal $\hat{G_c}$ should maximally reflect the information contained in $Y$. This supervision loss is formalized as:
> $\mathcal{L_s} = -\sum_{y \in \mathcal{Y}} p(y \mid G) \log p(y \mid \hat{G_c}), \quad \text{where } p(y \mid \hat{G_c}) = f_y(\hat{G_c}) \text{ and } \hat{G_c}= g(G),$
> which acts as a supervision loss for $\hat{G_c}$. By aligning the generated subgraphs $\hat{G_c}$ with the label $Y$, this loss implicitly accommodates scenarios where $P(Y \mid X)$ shifts between training and testing distributions, as it ensures that the invariant subgraphs capture features that are predictive of $Y$ across domains.
>
> Thus, the supervision loss in our framework satisfies the requirements for addressing concept shift, as it optimizes $\hat{G}_{c}$ to retain the key features that correlate with $Y$ even when $P(Y \mid X)$ changes. This design ensures that our framework remains effective under both covariate and concept shift scenarios, providing robustness in OOD generalization tasks.
>
> We appreciate the opportunity to clarify this point and have included this explanation in the revised manuscript to better demonstrate the theoretical support for handling concept shift scenarios.
>
>
> ---
>
> ### **2. Handling Inductive Learning Cases**
>
> >  **Comment:** While the influential node extraction method appears reasonable for transductive learning scenarios, the paper lacks clear explanation for handling inductive learning cases like molecular property prediction, where only the input graph is available without access to other domain graphs. Although the authors mention using nodes within input graph $\mathcal{G}$  to learn domain-specific representations $H$ for graph-level tasks, exploring the use of virtual nodes could be an interesting direction as they might better capture domain-specific information without relying on the existing node set.
>
>
> **Response:** Thank you for your observation regarding the limitations of the influential node extraction method in inductive learning scenarios. We would like to clarify that our design is also applicable in the inductive learning setting. When only the input graph is available, the selected influential nodes are derived directly from the input graph itself, rather than from any external graphs. This ensures that the method remains feasible and effective in inductive learning scenarios.
>
> Meanwhile, we agree that the use of virtual nodes could offer an interesting and potentially effective approach for capturing domain-specific information. We will consider exploring such designs in future work to further enhance the capabilities of our framework. Thank you for the precious suggestion.
>
> ---
>
> ### **3. Requested Changes and Typos**
> >  **Comment:** 'its' should be 'it'. 'unaviable' should be 'unavailable'
>
> **Response:** Thank you so much for your advice. We have accordingly modified the typos in our revised version.
>
> ---
>
> Thank you again for your thoughtful and constructive feedback. These suggestions have helped us identify areas for improvement, and we are committed to addressing them in the revised manuscript.
>
> Best,
> The Authors

---

> ### Comment · Reviewer_EFEr · 2025-01-10
>
> Dear Authors,
>
> I apologize for my delayed response in providing this feedback. Thank you for your detailed explanations in the rebuttal. I have carefully reviewed your responses addressing the initial concerns, and I have taken these clarifications into consideration while making my final recommendation. Your thorough responses have been very helpful in completing my evaluation of the paper.
>
> Best regards,
> Reviewer EFEr

---

> ### Author Response · Authors · 2025-01-10
> **Response to Reviewer EFEr**
>
> Dear Reviewer EFEr,
>
> Thank you for your follow-up and for taking the time to review our response. Your insights have been invaluable in refining and improving our work.
>
> Thank you again for your efforts.
>
> Best,
> The Authors

---

### Review · Reviewer_pCdy · 2024-11-28

**Summary Of Contributions:**

This paper introduces Generative Risk Minimization (GRM), a novel framework for handling out-of-distribution (OOD) generalization on graphs. The contributions include:
- Instead of extracting subgraphs, GRM generates invariant subgraphs in a continuous manner.
- The framework preserves the same set of nodes as the input graph while having continuous edge weights and node representations.
- A theoretical analysis that ensures the effectiveness of the generated subgraphs.

**Audience:**

Yes

**Claims And Evidence:**

Yes

**Requested Changes:**

- It is not intuitive to understand that the domain-related information pertains to the 1-hop neighbors of each node. Could the author include some visualizations (or other illustrations) to validate this hypothesis?
- Could the author provide the training time for larger graphs (Elliptic&Arxiv) to better reflect the time complexity?
- For the ablative studies, could the author provide insights on why certain components work better?

**Strengths And Weaknesses:**

**Strengths**
- SOTA performance on widely-used benchmarks.
- Works for both node-level and graph-level tasks.
- Provides theoretical justification.

**Weaknesses**
- Presentation needs to be improved. For example, there are some typos, and the transition from equation (8) to equation (9) is not very straightforward. Omitting the derivation makes comprehension more difficult.
- It is not intuitive to understand that the domain-related information is related to the 1-hop neighbors of each node.
- The artificial distribution shift experiments (Cora-Mix) are simple and may not be general enough. Different types of shifts are less explored.

---

> ### Author Response · Authors · 2024-12-24
> **Response to Reviewer pCdy (1/2)**
>
> Dear Reviewer pCdy,
>
> We sincerely thank you for your thoughtful review and valuable feedback on our work. Below, we would like to address and weaknesses mentioned and provide responses to the requested changes.
>
>
> ---
>
> ### **1.Presentation Improvements:**
>
> > **Comment:** "Presentation needs to be improved, e.g., typos and unclear transition from equation (8) to equation (9)."
>
> **Response:** Thank you for pointing this out. We have carefully reviewed the manuscript to correct any typos and ensure the transition between Eq. (8) and Eq.(9) is clearer. Specifically, we have added a more detailed description after Eq. (9) to aid comprehension.
>
> ---
>
>
> ### **2. Domain-Related Information and 1-Hop Neighbors:**
> >  **Comment:** "It is not intuitive to understand that domain-related information pertains to the 1-hop neighbors of each node."
>
> **Response:** We agree that this aspect requires further elaboration. To address this, we would like to clarify our decision as follows.  We select the 1-hop neighbors as the domain-related information because they are the most influential nodes for the specific node $v_i$ (i.e., the central node). According to Theorem 1 (Decaying Property of Node Influence) in [1], the influence of one node on another decreases **exponentially** as the distance between the two nodes increases. As a result, to determine the nodes from the domain that are most influential for a specific node, it is logical to prioritize nodes with the smallest distances. This justifies our choice of 1-hop neighbors as the most relevant domain-specific information. We have updated the manuscript to include this rationale in the revised version.
>
>
> > Theorem 1 (Decaying Property of Node Influence) Let $t$ be a path between node $u$ and node $v$ and let $D_{\text{GM}}^t$ be a geometric mean of node degrees occurring on path $t$. Let $D_{\text{GM}}^* = \min_t \{ D_{\text{GM}}^t \}$ and $h_* = d(u, v)$. Consider the node influence $I_{u,v}$ from $v$ to $u$. Then, $I_{u,v} \leq \frac{C}{(D_{\text{GM}}^*)^{h_*}}$, where $C$ is a constant.
>
> Ref: [1] Huang, Kexin, and Marinka Zitnik. "Graph meta learning via local subgraphs." Advances in neural information processing systems 33 (2020): 5862-5874.
>
>
> ---
>
>
> ### **3. Artificial Distribution Shifts:**
> > **Comment:** "The artificial distribution shift experiments (Cora-Mix) are simple and may not be general enough. Different types of shifts are less explored."
>
>
> **Response:** We would like to point out that although the artificial distribution shifts are straightforward, this design is the first to introduce a controllable degree of distribution shifts among existing methods. This approach provides a comprehensive view of the performance drop of various methods under increasing distribution shifts.
>
> Moreover, we have included different types of shifts in the six datasets used in our experiments. Specifically, we considered three types of shifts: (1) Artificial Transformations, (2) Cross-Domain Transfers, and (3) Temporal Evolution, as outlined in Table 1.
>
> In summary, we believe that the artificial distribution shift experiments offer a novel perspective for evaluating existing methods. Additionally, we have ensured that our experiments incorporate diverse types of shifts to comprehensively assess their impact.

---

> ### Author Response · Authors · 2024-12-24
> **Response to Reviewer pCdy (2/2)**
>
> ---
>
>
> ### **4. Training Time on Larger Graphs:**
> >  **Comment:** "Could the author provide the training time for larger graphs (Elliptic & Arxiv) to better reflect the time complexity?"
>
>
> **Response**: Thank you for the suggestion. We have provided the time/memory of all experiments conducted on a NVIDIA A6000 GPU with 48GB of memory. The results are as follows, and we have included them in Appendix C.3 in the revised version.
>
> | Dataset   | Total Training Time (s) | Time Per Epoch (s)| Memory Usage (MB) |
> |-----------|------------|-------|-------------------|
> | Cora      |  1,782.96   |3.91         |  1,329           |
> | Photo     |     3,504.65   |24.50       | 6,255               |
> | Twitch    | 1,156.37    |7.93          | 3,011              |
> | FB-100    |   5,902.7  |44.05        | 30,387       |
> | Elliptic  | 539.43  |16.19         | 4,103              |
> | Arxiv     |   4,355.21  |4.50       | 5,505             |
>
> From the results, we observe that all training times and memory usages are within a controllable range, demonstrating that our method is applicable to a wide variety of scenarios with different types of distribution shifts. Moreover, the training time per epoch is particularly higher for Elliptic, as graphs in this dataset contain significantly larger numbers of nodes and edges. Nevertheless, the total training time remains reasonable, indicating that our method is scalable to large datasets.
>
>
>  **Response:** Thank you for the suggestion. We will add a table summarizing the training time of GRM on larger datasets, such as Elliptic and Arxiv. This will provide a clear understanding of the time complexity and computational feasibility of our framework.
>
>
> ### **5. Ablative Studies:**
> >  **Comment:** "For the ablative studies, could the author provide insights on why certain components work better?"
>
> **Response:** We appreciate this suggestion and have expanded our discussion of the ablation studies. Specifically, we added deeper insights into how and why certain components (e.g., the regularization loss) contribute to the performance of GRM, in Section 4.4 in our revised version.
>
>
> ---
>
> Thank you again for your constructive feedback. We believe that these revisions will significantly improve the quality and clarity of our work, and we appreciate the opportunity to address your concerns.
>
> Best,
> The Authors

---

> > ### Comment · Reviewer_pCdy · 2024-12-30
> >
> > Dear Authors,
> >
> > Thank you for your further explanation. I have taken your responses into consideration in my final recommendation.
> >
> > Best,
> > Reviewer pCdy

---

> > > ### Author Response · Authors · 2024-12-30
> > > **Reponse to Reviewer pCdy**
> > >
> > > Dear Reviewer pCdy,
> > >
> > > Thank you for your thoughtful consideration of our responses. We appreciate the time and effort you have dedicated to reviewing our work and providing valuable feedback. We are more than happy to address any further concerns.
> > >
> > > Best,
> > > The Authors

---

### Decision · Action_Editor_pY1y · 2025-01-17

**Recommendation:** Accept with minor revision

**Comment:**

The reviewers acknowledge that the proposed method advances OOD generalization on graphs by generating continuous-weighted invariant subgraphs, supported by rigorous theoretical analysis. Despite some limitations in training efficiency for large graphs and questions about artificial distribution shifts in experiments, the work represents a substantial contribution to the field with its strong theoretical foundations and practical utility. However, there are still some concerns regarding the experimental settings and ablation studies. Thus, the authors should revise the manuscript based on the reviewers' comments.

**Audience:**

The researchers on out-of-distribution detection and generalization will be interested in this work.

**Claims And Evidence:**

Yes, the proposed method is verified by sufficient experiments and comparisons.